# The Use of Radiomic Tools in Renal Mass Characterization

**DOI:** 10.3390/diagnostics13172743

**Published:** 2023-08-24

**Authors:** Beatriz Gutiérrez Hidalgo, Juan Gómez Rivas, Irene de la Parra, María Jesús Marugán, Álvaro Serrano, Juan Fco Hermida Gutiérrez, Jerónimo Barrera, Jesús Moreno-Sierra

**Affiliations:** 1Department of Urology, Clínico San Carlos Hospital, Health Research Institute of Clínico San Carlos Hospital, Complutense University, 28040 Madrid, Spain; irn_parra@hotmail.com (I.d.l.P.); chusi.marugan@gmail.com (M.J.M.); alvaro.serrano.p@gmail.com (Á.S.); jfhermidag@gmail.com (J.F.H.G.); dr_jmoreno@hotmail.com (J.M.-S.); 2Radiodiagnosis Department, Clínico San Carlos Hospital, Complutense University, 28040 Madrid, Spain

**Keywords:** benign, malignant, kidney, tumor, radiomics, renal mass, renal cell carcinoma, texture analysis, prognosis

## Abstract

The incidence of renal mass detection has increased during recent decades, with an increased diagnosis of small renal masses, and a final benign diagnosis in some cases. To avoid unnecessary surgeries, there is an increasing interest in using radiomics tools to predict histological results, using radiological features. We performed a narrative review to evaluate the use of radiomics in renal mass characterization. Conventional images, such as computed tomography (CT) and magnetic resonance (MR), are the most common diagnostic tools in renal mass characterization. Distinguishing between benign and malignant tumors in small renal masses can be challenging using conventional methods. To improve subjective evaluation, the interest in using radiomics to obtain quantitative parameters from medical images has increased. Several studies have assessed this novel tool for renal mass characterization, comparing its ability to distinguish benign to malign tumors, the results in differentiating renal cell carcinoma subtypes, or the correlation with prognostic features, with other methods. In several studies, radiomic tools have shown a good accuracy in characterizing renal mass lesions. However, due to the heterogeneity in the radiomic model building, prospective and external validated studies are needed.

## 1. Introduction

Due to improvements in diagnostic techniques, the incidence of renal mass detection has increased during recent decades. Some series have shown that around 20–30% of small renal masses (<4 cms) are benign tumors [1,2,3].

To avoid unnecessary surgeries, there is an increasing interest in clinical and radiological features that could help to predict histological results [4,5]. In recent years, many tools have been potentiated to enhance image interpretation, including radiomics and artificial intelligence (AI) [6,7]. However, currently, no predictive models are available in clinical practice [4].

AI, as a concept, refers to the use of computational technologies to make predictions simulating intellectual procedures typical of human intelligence [4,8]. Many applications of this computer science branch are being studied [8,9]. Radiomics is a tool that converts medical images into quantitative features and data that can help in decision support. This can solve the actual limitations of conventional imaging tests [4]. A common tool of radiomics is texture analysis, which allows the quantification of heterogeneity in the region of interest in the image, and the correlation of these dates with clinically relevant variables [10].

## 2. Radiomics and Texture Analysis

Radiomics in renal cancer can be used in lesion characterization, and to assess the malignance probability before a treatment decision assessment and therapy response [10]. Lesion characterization can help to correlate the imaging features with different tumor characteristics. These tumor characteristics can be related to the histological data, tumor grade, or genetic and molecular patterns [4].

One of the most important tools used is texture analysis. This tool uses the pixel distribution of images, especially from magnetic resonance (MR) imaging and computed tomography (CT). The extrapolation of the quantitative data from the imaging tests cannot be achieved through only human interpretation [4]. These data can later be used to build algorithms that allow the differentiation of tissue characteristics, which can help, for example, in the differentiation of tumor lesions between benign and malignant tumors [11].

Texture analysis assesses the pixel and voxel gray-level distribution in a region of interest (ROI) in the image; subsequently, the distribution is represented by a histogram. There are different ways of performing texture analysis, with the most frequent being the use of statistics-based techniques [10]. The main variables used in statistical analysis are the gray-level intensity, asymmetry, kurtosis, entropy, and mean of the positive pixels for each histogram [12]. Figure 1 shows a summary of how the texture analysis is performed.

After image features analysis using radiomics, it is possible to use AI (via machine learning and deep learning algorithms) to analyze these data, and create models that can help in treatment decisions [11].

## 3. Radiomics in Renal Mass Characterization

Different kidney lesions can vary between malignant and benign tumors, and the differentiation of these main groups is essential to making a correct treatment decision. Although MR and CT are used to assess tumor characteristics and help to distinguish between benign and malignant, there are new tools based on radiomics that can give more information [4,13].

There are many lesions, including renal cell carcinoma (RCC), that are low in attenuation; in these cases, heterogeneity is an important feature to analyze. This heterogeneity can be assessed objectively, using texture analyses [10,14,15].

Varghese and collaborators [16] analyzed, retrospectively, 129 cases of RCC and 45 cases of benign renal masses, angiomyolipomas (AMLs), and oncocytomas, using texture analysis from multiphase CT scans. Whole lesion regions of interest (ROIs) were established. The entropy, entropy of fast-Fourier transform magnitude, mean, uniformity, information measure of correlation 2, and sum of averages were the metrics used to calculate the area under the curve (AUC) to differentiate between benign and malignant masses, ranging from 0.80 to 0.98 depending on the combinations. The texture model with all texture metrics had an AUC of 0.87 (*p* < 0.05).

Wentland et al. used a machine learning (ML) model in 148 renal masses evaluated via CT scan, to differentiate between benign and malignant lesions. The features were selected using the random forest model. Their ML model achieved an overall accuracy of 0.82, and an AUC of 0.80 [17].

Recently, Feng and collaborators performed, retrospectively, an analysis of 156 small renal masses (92 malignant tumors vs. 64 benign tumors), using radiomics. From the CT scan, they drew three-dimensional ROIs, to extract radiomic features. The tumors were classified into three categories: A (AML with visible fat), B (benign SRM without visible fat, and C (malignant SRM). After selecting the optimal features, they elaborated three models: the clinical factor model, radiomics signature, and radiomics nomogram. In the differentiation between categories B and C, the radiomics nomogram showed the highest AUC, but no significant differences were found between the three models (*p* > 0.5). The radiomics nomogram showed a better discrimination when comparing clinical factors to differentiate between benign and malignant lesions (*p* = 0.0007) [18].

Due to the more frequent use of CT scans in general clinical practice, there are not many studies using texture analysis with MR. Hoang et al. [19] evaluated the quantitative texture parameters in 142 renal lesions characterized by multiphasic contrast-enhanced MR; 90 were clear cell carcinomas, 22 were papillary carcinomas, and 30 were oncocytomas. A histogram was used to differentiate benign and malignant lesions and texture imaging features, to differentiate the carcinoma subtype. It showed a good specificity when comparing benign and malignant lesions (oncocytomas versus RCCs: 85.8%).

One of the largest series using MR-based models included a total of 1161 renal lesions (655 malignant and 507 benign), and was a study performed by Xi and collaborators [20]. They compared the results for the differentiation of benign and malignant lesions between a deep learning (DL) model, expert-based model, and radiomics. The best results were found with the DL model, with an accuracy of 0.70.

Xu et al. used deep learning to differentiate between benign and malignant tumors. They found a good AUC using DL models, working from MR images, based on T2-weighted images, diffusion-weighted imaging (DWI), and a combination of both. The AUCs were 0.906, 0.846, and 0.924, respectively; the AUC was highest using the model combination [21].

A recent study performed by Massa’a et al. [22] evaluated 182 renal masses with ML models, using features extracted from MR. The best accuracy was found using features from T2-weighted images. The three best-performing ML models were the support vector machine (SVM), logistic regression (LR), and linear discriminant analysis (LDA), with an AUC of 0.79, 0.78, and 0.77, respectively, without statistically significant difference, between these models (*p* > 0.5).

### 3.1. Angiomyolipoma

AMLs are benign tumors that have an important fat component. However, some of them have a minimal fat component, and these cases are difficult to differentiate from RCCs [10,23]. Several studies have compared image characteristics between AMLs and RCCs [23,24,25,26,27,28]. Some of them are summarized in Table 1.

In 2016, Takahashi et al. [26] compared the fat presence in 38 AMLs and 75 RCCs by comparing subjective evaluation with objective methods using CT pixel distribution and texture analysis. Macroscopic fat was identified in 15/38 AMLs and 1/83 RCCs. Through the addition of CT negative pixel distribution into the texture analysis, fat was identified in 20/38 AMLs and 1/83 RCCs, but this difference from the subjective method was not statistically significant.

Feng et al. [27] used the ML-based texture analysis of CT images to differentiate AMLs from RCCs in 58 patients with small renal masses <4 cm (17 AMLs vs. 41 RCCs). To establish discriminative classifiers, SVM was used. Of the 45 features extracted from the images, 16 showed significant intergroup differences. Selecting the optimal 11 features, and using SVM, obtained the highest AUC in differentiating fat-poor AMLs (up to 0.955). These 11 features were: mean, median, 10th percentile, 25th percentile, 75th percentile, 90th percentile, skewness, entropy (these first in the unenhanced phase), 25th percentile (in the corticomedullary phase [CMP]), energy, and entropy (in the nephrographic phase).

Cui EM et al. [23], performed a retrospective study of 171 renal masses, using texture analysis with machine learning, to better discriminate AMLs from other lesions. They extracted image features from the whole tumor in three phases of CT, differently to other studies that extracted these features from different tumor slices. They found that ML performed best in differentiating AMLs from all RCCs, with an accuracy of 92.69%. The most important findings to differentiate these cases were the hyperattenuation and homogeneity of the pre-contrast phase. On the other hand, they showed the worst results when differentiating AMLs from non-RCCs, probably because these lesions can have a similar appearance to AMLs (homogeneous and hyperattenuating). They found better results using computer-assisted methods, when comparing the same cases with the subjective morphological features recognized by radiologists.

Wang and collaborators [28] evaluated different selected ROIs to assess their impact on differentiating AMLs from RCCs. They evaluated, retrospectively, 31 patients with AMLs and 74 with RCCs. They analyzed different quantitative parameters determined by three different ROIs. ROI 1 and ROI 2 were placed in the part of the tumor that showed more enhancement, varying the area size (smaller for ROI 2), and ROI 3 was placed on the largest image of the tumor, to measure the heterogeneous degree. All these parameters were evaluated via a four-phase CT scan. They observed that the heterogenous degree determined by ROI 3 was higher for RCCs than AMLs. In the evaluation of the enhanced attenuation value, the parameters determined by ROI 2 showed a better diagnostic performance. The ROC curves calculated with the parameters during the CMP were higher [28].

### 3.2. Oncocytoma

An oncocytoma is a benign renal neoplasm, with a prevalence that varies between 3–7%, when evaluating small renal masses (<4 cms) [29]. It is shaped by polygonal and eosinophilic cells that are rich in mitochondria. These cells are similar to chromophobe RCCs, which can complicate their differentiation, because of similar image findings and pathology characteristics [10].

Due to the differences in treatment and prognosis between these two entities, the interest in tools that allow us to improve their diagnostic has increased. Many articles have studied radiomics in terms of distinguishing these renal lesions.

In the cohort of patients analyzed by Varghese et al. [16], 27 cases of renal cell oncocytoma were evaluated. They found that adding a Fourier analysis to the histogram analysis significantly improved the AUC in 0.20. The full texture model had an AUC of 0.90 (*p* < 0.05) for differentiating oncocytomas from the rest of the tumors in the study.

Hoang et al. [19] elaborated three different models using texture analysis, with parameters obtained from multiphase MR. Analyzing oncocytomas, the models showed a good accuracy in distinguishing oncocytomas from malignant lesions (oncocytomas versus RCCs: 77.9%; oncocytomas versus clear cell RCCs [cc-RCCs]: 79.3%). Due to the lack of diagnoses of chromophobe histology tumors in their center, the model could not be calculated for this subtype of RCCs.

More recently, Uchida et al. [30] analyzed, retrospectively, a 49-patient cohort with renal masses diagnosed via 1.5T MRI. After partial or radical nephrectomy, the pathological anatomy results were 41 chromophobe RCCs (ChRCCs) and 8 renal oncocytomas (ROs). They obtained image features via MR image and texture analysis, using apparent diffusion coefficient (ADC) maps. They obtained 49 texture features, with eight of them contributing significantly to the differentiation between ChRCCs and ROs. The mean ADC value was the one with more relevance in differentiating these two entities. Their results showed that the ChRCCs had a lower mean ADC value than the ROs (*p* < 0.0001).

### 3.3. Subtypes of Renal Cell Carcinoma (RCC)

As is known, due to the increase and improvements in imaging techniques, an increase in the diagnosis of renal masses has been observed. The main problem is the risk of overtreatment in 20–30% of resected renal masses, which are benign [1]. Besides, the most common malignant tumor is the RCC, which includes different subtypes that have different morphological and pathological features, due to their different biological behaviors, which can lead to different prognoses [31]. For example, in low-grade cc-RCCs, minimally invasive surgery could be planned safely, but in high-grade tumors, precaution should be taken [32]. Additionally, the different biological behaviors among these tumors could help in the development of targeted therapeutic agents [33].

Kocac B et al. [33] elaborated a texture analysis model using CT images to distinguish different subtypes of RCC. They selected, retrospectively, 68 cases of RCCs (48 cc-RRCs, 13 pc-RCCs, 7 chc-RCCs), and added 26 RCC cases from public databases, to perform external validation (13 cc-RCCs, 7 pc-RCCs, 6 ch-RCCs). The features were extracted from unenhanced and CMP CT. Using the selected features, algorithms were elaborated. The base classifiers used were artificial neural networks (ANNs) and SVM. They obtained the best results in the discrimination of RCCs vs. non-RCCs with the CMP-image using ANNs, with poor results in distinguishing the main three RCC subtypes. To improve the reproducibility and generalizability, additional algorithms (such as adaptive boosting or bagging) can be used in machine learning.

As has been mentioned, Varghese and collaborators [16], after analyzing, retrospectively, 129 cases of RCCs and 45 cases of benign renal masses (AMLs and oncocytomas), using texture analysis from multi-phase CT scans, found that adding texture analysis methods to the base model resulted in a significant improvement in the AUC to differentiate the three subtypes of RCCs (clear cell, papillary, chromophobe).

Some studies evaluated the role of radiomics in differentiating subtypes of papillary RCCs due to their different behavior. Vendrami CL et al. [34] used texture analysis in 47 papillary RCC tumors characterized via MR images; 31 were type 1, and 16 were type 2. They performed a statistical model with qualitative and quantitative analyses, including 2D and 3D analysis. Significant differences showed in some MR image features; namely, the necrosis, heterogeneous enhancement, perilesional stranding, and metastases, which were more frequent in papillary type 2 RCCs. They found that this model, with 2D texture analysis in the VIBE sequence, showed a good prediction of type 2 tumors (AUC value of 0.87). Lastly, 3D analysis did not improve the model, compared to 2D analysis.

Duan C et al. [35] performed a single institution study evaluating the difference between the papillary renal cell carcinoma (PRCC) subtypes, using texture analysis. They analyzed, retrospectively, 62 patients with renal tumors, diagnosed via three-phase CT scans. Based on the pathology results, they included 30 type 1, and 32 type 2, PRCC cases. The features were extracted from ROIs that included the whole tumor. They saw that the entropy and correlation were increased in type 2 PRCCs, and that the best results in differentiating both types were obtained via the nephrographic-phase model, and the model using the texture parameters of three-phase CT.

### 3.4. Identification of Aggressive Tumor Features and Sarcomatoid Differentiation

The nuclear grade, represented by Fuhrman grading, is one of the most important and independent prognostic factors [36,37]. To avoid tumor biopsy due to a lower concordance rate with the Fuhrman grade (from 30–97%), the use of radiomics models to predict the nuclear grade has been studied [33,34,38].

Ding and collaborators [39] presented a cohort of 92 cases of cc-RCCs studied via CT images, compared with a validation cohort. They performed three different models, finding that adding texture features increased the accuracy of the prediction models for grading cc-RCCs in the study and validation cohort.

In 2019, another study performed by Lin and collaborators [32] assessed the impact of a machine learning model to predict the Fuhrman nuclear grade in a total of 232 cc-RCC lesions. Their results suggested that the classifier based on three-phase CT images showed an acceptable performance in predicting the Fuhrman grade, and was superior to single-phase CT images, with an AUC of 0.87.

Sarcomatoid RCCs are associated with aggressive behavior and a poor prognosis, with approximately 60–80% of patients having advanced or late-stage disease. For this reason, observation, ablative therapies, and nephron-sparing surgeries are not recommended, as a radical treatment is needed [40,41]. It is difficult to make a diagnosis before treatment, due to the heterogeneous distribution of the sarcomatoid regions, with the poor biopsy results being positive in approximately 7.5% of cases [42]. Some studies have valued the accuracy of radiomics in differentiating sarcomatoid histology [15,43,44].

In 2015, Schieda and collaborators [43] applicated texture analysis to differentiate 20 sarcomatoid RCCs and 25 cc-RCCs. All patients were studied via a CT scan. For the texture analysis, they extracted the gray-level co-occurrence and run-length matrix, showing that sarcomatoid RCCs were significantly more heterogeneous than RCCs (with a greater run-length nonuniformity and greater gray-level nonuniformity in sarcomatoid RCCs). When combining textural features, the AUC in the identification of sarcomatoid RCCs was 0.81 +/− 0.08 (*p* < 0.0001).

More recently, Meng et al. [44], compared 29 sarcomatoid RCCs with 99 cc-RCCs. After collecting 1029 different features extracted from the CMP and nephrographic phase (NP) via CT scans, they used the selected features to build different models, using radiomics approaches. The AUC value using the selected CMP and NP radiomics features was significantly higher than that using subjective characteristics. A better value of AUC was obtained using the NP nephrographic model, which could have an important role in differentiating sarcomatoid RCCs and cc-RCCs.

## 4. Discussion

AI is a branch of computer science that is attracting a continuous growing interest in the medical fields [45]. Just as it is being used for treatment aspects (for example, as an important auxiliary tool in surgery), it has a large potential in diagnostics. The use of AI can improve diagnostic efficiency. Its application in several diagnostic fields has been studied: radiological, pathological, ultrasonographic, endoscopic, and biochemical. Different departments are evaluating this tool, such as as pneumology, ophthalmology, cardiology, etc. [46]. In urology, there are many fields in which AI is being studied, such as urolithiasis, the benign enlargement of the prostate, pediatric urology, renal transplant, and prostate cancer, among others. It has a lower impact on less-frequent tumors, such as penile and testicular tumors [45]. Regarding renal masses, radiological assessment is essential, and the development of techniques that can improve diagnostic efficiency are growing. The interest in radiomics for improving the characterization of tumor lesions without invasive techniques has increased. Due to the different prognosis in some histological subtypes, and the low accuracy in the pathology results in some series of renal biopsies, radiomics is emerging as a powerful tool for renal mass characterization, the recognition of prognostic features and, moreover, helping with therapeutic decisions.

Nowadays, conventional images, such as CT or MR, are the most common diagnostic tools for renal mass characterization. Some features on a contrast-enhanced CT can suggest the tumor’s histology but, currently, there is no radiographic feature that can accurately predict histologic analysis. For example, a renal mass with a central stellate region, with a low level of enhancement, can suggest the diagnosis of an oncocytoma, which can be confused with central necrosis from an RCC [6]. Distinguishing between benign and malignant tumors in small renal masses is challenging using conventional images and, sometimes, the interobserver differences are considerable [43]. To improve subjective evaluation, and avoid conventional imaging limitations, the interest in using radiomics to extract quantitative parameters from medical images has increased.

The main problem with the studies evaluating radiomics is that are based on retrospective cases, with the limited evidence that comes with this. Prospective studies with a larger sample size are needed. Another limitation is the heterogeneous performance in the radiomic techniques, with different selected features, different calculations of ROIs, and different methods of performing texture analysis. This heterogeneity makes the assessment of external validation difficult. This means that none of the multifactorial algorithms have achieved clinical use, or been independently validated [47].

External validations are needed in order to achieve future clinical applications, with large-size and probably multicentric studies [48]. Of the mentioned papers, only one performed an external validation of radiomic models based on CT scans to distinguish between benign and malignant lesions, with some limitations, especially the small study size [33].

Most of the mentioned studies have been based on CT images, which is not uncommon, as it is used most frequently. Only a few studies have used features extracted from MR in their analysis, which can complicate external validation [34].

Another important characteristic that varies between studies is the selection of ROIs. Some cases select some slices of the tumor; other cases take the whole tumor volume. Some include hemorrhagic regions or calcifications, while other exclude these features. The different sizes of ROIs can affect the role of the quantitative parameters, which means that it is challenging to use the selection criteria in routine diagnoses and external reproducibility [28].

Moreover, the CT phases used to extract the different features are not constant. Some studies used single-phase, and others multi-phase CT. In some cases, there was a better accuracy among the models performed with the nephrographic phase [24,44], and in some, with CMP [28,33]. More studies are needed in order to assess the impact of different phases in radiomic models.

In a nutshell, there is an important heterogeneity in the radiomic features used in the different studies, with a lack of a standard selection of features that could be used to calculate uniform models. This would bring radiomics closer to clinical use [7].

## 5. Future Directions

The increase in the investigation of the role of radiomics in renal mass characterization shows that it is a good tool for discriminating between benign and malignant tumors, and for discriminating, in some cases, subtypes of RCCs, and evaluating imaging features that correlate with prognostic factors. To access this technology, and use AI, it is important to build high-quality datasets.

Prospective and multicenter studies are needed, in order to perform reproducible and homogeneous models with independent validation, to bring radiomics to clinical practice. 

Another field that is receiving increasing interest is radiogenomics, which is using radiomic features to correlate with prognostic and predictive biomarkers, and anticipate the treatment response. The associations between quantitative features and mutational genes or molecular markers are being investigated [4,6,45].

## 6. Conclusions

Several applications of AI are being investigated regarding their use in clinical practice. Due to the high incidence of benign tumors in SRM, and to avoid overtreatment, the use of radiomics to achieve better accuracy in the evaluation of these tumors is being studied. Radiomic tools have shown good accuracy in characterizing renal mass lesions in several studies, proving useful in most series in distinguishing between different types of renal tumors, and between benign and malign tumors, and in the detection of radiological features associated with a bad prognosis. However, due to the heterogeneity in the technique methods, and the different algorithms used, prospective studies with homogeneous algorithms are needed, to assess the real impact of using this tool in clinical assistance.

## Figures and Tables

**Figure 1 diagnostics-13-02743-f001:**
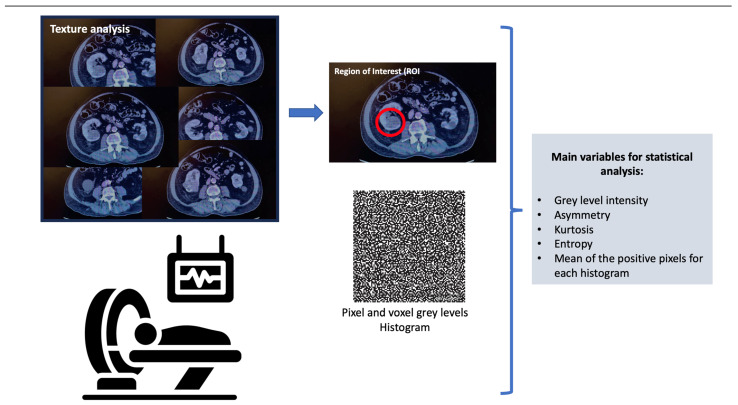
Texture analysis.

**Table 1 diagnostics-13-02743-t001:** Evaluation of AMLs via texture analysis.

	Evaluation of AMLs via Radiomics			
Study	Objetive	Study Design	Model	Results
**Yan L et al. [25]** ** *Year 2015* **	Discrimination of AMLs with minimal fat, ccRCCs, and pRCCs	- Patients evaluated via CT scan - 18 AMLs vs. 18 ccRCCs vs. 14 pRCCs - Retrospective study	- Texture analysis	Classification accuracy: - Up to 100% for minimal-fat AMLs, vs. ccRCCs with PCP - 100% for minimal-fat AMLs, vs. pRCCs with NP - 97.5% for ccRCCs, versus pRCCs with CMP
**Hodgdon et al. [24]** ** *Year 2015* **	Accuracy of texture analysis to differentiate fat-poor AMLs from RCCs	- Patients evaluated via CT scan - 16 fat-poor AMLs vs. 84 RCCs - Retrospective study	- Texture analysis - SVM (based in most discriminative features)	- Lower homogeneity and higher entropy in RCCs (≤0.1) - Model with several texture features resulted in an AUC of 0.89 ± 0.04. - Difference in AUC between textural features and subjective visual heterogeneity was 0.25 (*p* = 03)
**Takahashi et al. [26]** ** *Year 2016* **	Detection of fat presence using pixel distribution and texture analysis.	- Patients evaluated with CT scan - 38 AMLs and 83 RCCs - Retrospective study	- CT pixel distribution - Texture analysis	- Fat was identified in 15/38 AMLs and 1/83 RCCs using subjective methods and CT negative pixel distribution - Combining CT negative pixel distribution with texture analysis, fat was identified in 20/38 AMLs
**Feng et al. [27]** ** *Year 2018* **	Differentiate fat-poor AMLs from RCCs in small renal masses (<4 cm)	- Patients evaluated via CT scan - 17 AMLs vs. 41 RCCs - Retrospective study	- Texture analysis - SVM (ML)	- SVM-RFE classifier with selected 11 optimal features obtained the highest AUC in differentiating fat-poor AMLs (up to 0.955)
**Cui EM et al. [23]** ** *Year 2019* **	Differentiate fat-poor AMLs from RCCs	- Patients evaluated via CT scan (three phase images: PCP, CMP, NP) - 41 AMLs vs. 130 RCCs (82 ccRCCs, 22 pRCCs, 26 chRCCs) - Retrospective study	- Texture analysis - SVM (ML)	- ML based in whole tumor CT texture features (from three-phase image) showed the best accuracy in differentiating AMLs from all RCCs (accuracy of 92.69%)
**Wang et al. [28]** ** *Year 2021* **	Differentiate AMLs without visible fat from ccRCCs	- Patients evaluated via CT scan - 31 AMLs vs. 74 cc-RCCs - Retrospective study	- Texture analysis (Three different ROIs calculated)	- The heterogenous degree determined by ROI 3 was higher for RCCs than AMLs - Images from CMP had the highest diagnostic performance

Abbreviations: AML, angiomyolipoma; ccRCC, clear cell renal cell carcinoma; pRCC, papillary renal cell carcinoma; CT, computed tomography; PCP, precontrast phase; NP, nephrographic phase; CMP, corticomedullary phase; SVM, support vector machine; AUC, area under the curve; ML, machine learning; RFE, recursive feature elimination; ROI, region of interest.

## Data Availability

Not applicable to this article.

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
