# Peer review of "The Use of Radiomic Tools in Renal Mass Characterization"

_diagnostics, 2023, doi:10.3390/diagnostics13172743_

Round 1
Reviewer 1 Report
The authors reviewed the updated knowledge about the role of radiomics in renal mass characterization reported in the Literature. Their study regarded several clinical applications of texture analysis in benign and malign renal tumors, such as angiomyolipoma, oncocytoma, and different subtypes of renal cell carcinoma.
To improve the quality of the text and its usefulness to the readers, I would recommend only a few minor comments:
- Kindly explain all the abbreviations used in the text and Table 1. Please add a legend to Table 1.
- Please carefully review the references list according to the Diagnostics journal recommendation (e.g. the journal name - is in italics).
- The text is written in different theme fonts and sizes.
Author Response
Good afternoon.
-In the file attached you will find the article with the modified table and all the abreviations added. All the changes are in red.
-I have unificated the fonts and the size of the text and reviewed the references. All journal´s name are abreviated and with italics because of the the journal recommendations, it is that correct?
Thank you for your comments.

Reviewer 2 Report
I read with interest you work about the use of radiomics for the characterization of renal masses, but the article has some major issues. It is too short to be a review, the Literature it is not sufficiently and clearly summarized and the majority of articles are about texture analysis, that is only a limited part of radiomic. Furthermore, Authors didn't analyze articles of 2023 and 2022 (only 3-4 articles published in 2022), instead there are some interesting works in recent Literature such as:
Feng S, Transl Oncol 2023
Massa'a RN, Abdom Radiol 2022
Wentland AL, Abdom Radiol 2023
And so on.
Furthermore, some citations are incorrect: for example you cite the study of Takahashi (reference 22) in the text (line 104-108) and in Table 1, but the number of RCC it's not correct because you report at first 75 RCC and the 83 RCC: how it is possible? correct it.
-Table 1: you have to revise and rewrite the table because it is very unclear: not only the layout it is very poor (you change style and dimension of the text), but for each study you have to report the method of data acquisition (CT or MRI), and this datum it's not available for each article. Furthermore you have to summarize more precisely the model used and the result. The table in present form is not suitable for publication.
The quality of English is good, except for Table 1.
Author Response
Good afternoon.
In the file attached you will find the article with the revisions in red.
- I have redone the table and added the recent articles that you recommended.
- The review article is longer now with more references.
- Sorry about the confusion with Takahashi study. There were a total of 83 RCC in 75 patients, that was the confusing fact. I have corrected it.
Thank you for your comments.

Reviewer 3 Report
The review by Gutiérrez Hidalgo et al is well written with clear introduction and literature overview. However, some more graph or figures are needed to be added, as well as some other examples where AI is used in biomedicine with successfull aplications as well as the ones that did not make some relevant progress. The discussion part should add some more detail in this regard. The conclusion should be highlighted.
Author Response
Good afternoon.
In the file attached you will find the article with some modifications in red.
I have added a new figure about texture analysis that you can find at the end of the document.
I have reviewed de discussion and the conclusion part, with other uses of AI. You will find the modification in red.
Thank you for your comments.

Round 2
Reviewer 2 Report
I read with interest the corrections of the paper but, unfortunately, in my opinion this is not suitable for publication in Diagnostics. The title is catchy, because it states "radiomics tools in renal masses", but reading the article it is clear that Authors cite only articles about texture analysis (this is clear reading Table 1), that it's only a part of all radiomics tools, and furthermore it is the more old and not so useful in clinical practice because it requires in most of the cases to segment the diagnostic images. The only studies about machine learning and deep learning were added on my suggestion, but Authors didn't deepen this part of the Review.
The added parts in red are very generic (see the initial part of the Discussion) and English is too poor. There are still some English errors (line 276: "it's application") even in the "corrected" part.
Figure 1 is not meaningful. There is still a lack of well-done and significant Figures and Tables, that are required in every Review.
The quality of English is too poor regarding the quality of the sentences and the scientific soundness. There are some minor errors. There are many repetitive sentences.
Author Response
Dear Diagnostics editorial team,
Subject: Re: Reviewers replay on the Paper "The use of radiomic tools in renal mass characterization”
I hope this message finds you well. Thank you for taking the time to review our paper titled "The use of radiomic tools in renal mass characterization" submitted for consideration in Diagnostics. We appreciate your interest in our work and the valuable feedback you provided.
We appreciate the feedback from reviewers 1 and 3 accepting our manuscript.
We regret that the paper did not meet the expectations of reviewer 2. Your comments have been carefully considered, and we genuinely value your insights.
We acknowledge your concern about the title of the paper and the misrepresentation it may cause. We apologize for any confusion that may have arisen due to this issue. Upon reflection, we agree that the current title may not accurately encompass the entire scope of the review. Our intention was to cover a broader range of radiomics tools, including texture analysis, machine learning, and deep learning. The following references beyond texture analysis are included and discussed within the manuscript:
- Gurbani S, Morgan D, Jog V, et al. Evaluation of radiomics and machine learning in identification of aggressive tumor features in renal cell carcinoma (RCC). Abdom Radiol (NY). 2021;46(9):4278-4288
- Wentland AL, Yamashita R, Kino A, et al. Differentiation of benign from malignant solid renal lesions using CT-based radiomics and machine learning: comparison with radiologist interpretation. Abdom Radiol (NY). 2023;48(2):642-648.
- Massa'a RN, Stoeckl EM, Lubner MG, et al. Differentiation of benign from malignant solid renal lesions with MRI-based radiomics and machine learning. Abdom Radiol (NY). 2022;47(8):2896-2904.
- Feng Z, Rong P, Cao P, et al. Machine learning-based quantitative texture analysis of CT images of small renal masses: Differentiation of angiomyolipoma without visible fat from renal cell carcinoma. Eur Radiol. 2018;28(4):1625-1633.
- Lin F, Cui EM, Lei Y, Luo LP. CT-based machine learning model to predict the Fuhrman nuclear grade of clear cell renal cell carcinoma. Abdom Radiol (NY). 2019;44(7):2528-2534.
- Kocak B, Yardimci AH, Bektas CT, et al. Textural differences between renal cell carcinoma subtypes: Machine learning-based quantitative computed tomography texture analysis with independent external validation. Eur J Radiol. 2018;107:149-157.
- Suarez-Ibarrola R, Hein S, Reis G, Gratzke C, Miernik A. Current and future applications of machine and deep learning in urology: a review of the literature on urolithiasis, renal cell carcinoma, and bladder and prostate cancer. World J Urol. 2020;38(10):2329-2347.
- Jiang Y, Yang M, Wang S, Li X, Sun Y. Emerging role of deep learning-based artificial intelligence in tumor pathology. Cancer Commun (Lond). 2020;40(4):154-166.
- Xi IL, Zhao Y, Wang R, et al. Deep Learning to Distinguish Benign from Malignant Renal Lesions Based on Routine MR Imaging. Clin Cancer Res. 2020;26(8):1944-1952.
- Xu Q, Zhu Q, Liu H, et al. Differentiating Benign from Malignant Renal Tumors Using T2- and Diffusion-Weighted Images: A Comparison of Deep Learning and Radiomics Models Versus Assessment from Radiologists. J Magn Reson Imaging. 2022;55(4):1251-1259.
Furthermore, we appreciate your feedback about the language quality in the added portions of the paper. We apologize for the shortcomings in English proficiency. Moving forward, we have engaged with language editing services to improve the overall quality of the manuscript in terms of language and expression.
Regarding the figures and tables, we understand their importance in enhancing the clarity and significance of a review article. Your observation regarding Figure 1 and the lack of well-developed and informative figures and tables is valid. We have worked diligently to improve the visual representation of our findings and ensure that all necessary data are adequately presented.
Once again, we thank you for your valuable evaluation and critique. We hope the manuscript fulfils the standards of the journals in its new form.
Sincerely,
The authors
Reviewer 3 Report
The manuscript is now improved according to reviewers comments.
Author Response

(The authors gave the same response as above.)
